# DHKR: Dynamic Hierarchical Knowledge Routing for Efficient Low-Resource Alignment

## Abstract

Knowledge-Aligned Domain Shift Tuning (KADA) is a PEFT framework based on the Lottery Hedge Fund Hypothesis (LHFH) to identify and reuse latent knowledge fragments. Although KADA bridges knowledge gaps between source and target domains, it relies on a fixed set of subnetworks, which limits flexible adaptation and prevents automatic discovery of optimal model capacity. Existing MoE and dynamic PEFT methods lack a unified mechanism that jointly enables adaptive capacity growth and strong routing stability.

To address these limitations, DHKR employs a two-level routing mechanism to expand subnetworks hierarchically on demand (domain $\rightarrow$ modality), explore and adjust capacity ($K \times L$) as needed for new domains. To support dynamic capacity growth, DHKR stabilizes routing via a composite growth trigger—monitoring stagnation, entropy, imbalance, and instability—and multi-level Loss-Free Balancing (LFB). Ablation studies show that these mechanisms reliably prevent routing instability during growth. Like KADA, DHKR places the Knowledge Steering Layer (KSL) immediately below the LM head and inherits its heritage, enabling efficient parallel routing while keeping the 4-bit backbone frozen. Experiments show that DHKR improves calibration (ECE 0.02 vs. KADA 0.13) and lowers training cost (5.67 sec/iter vs. AdaLoRA 15.00 sec/iter), demonstrating both robustness and practical efficiency. DHKR provides a unified design for dynamic, knowledge-aligned adaptation for knowledge jackpots while maintaining routing and calibration stability.

## 1 Introduction

Large language models (LLMs) achieve strong general performance, but adapting them to specialized domains such as medicine or law often requires full-model fine-tuning, which is costly and inefficient. Parameter-Efficient Fine-Tuning (PEFT) methods, including adapters (Houlsby et al., 2019), LoRA (Hu et al., 2022), BitFit (Zaken et al., 2022), and QLoRA (Dettmers et al., 2023), update only a small subset of parameters, but most PEFT methods do not explicitly leverage the latent knowledge encoded during pretraining.

Knowledge-Aligned Domain Shift Tuning (KADA) (Kawamae, 2025) addresses this limitation by reusing latent knowledge fragments under the Lottery Hedge Fund Hypothesis (LHFH), which posits that task-relevant knowledge is concentrated in high-frequency components of model weights. In LHFH, knowledge is defined as patterns and reasoning structures learned during pretraining, rather than just a collection of facts, with hidden high-performance networks interpreted as 'winning lottery tickets.' However, KADA relies on a static, predefined set of subnetworks, which limits its ability to adapt flexibly or discover the optimal capacity. Existing MoE (Jordan & Jacobs, 1994; Shazeer et al., 2017; Fedus et al., 2022; Guo et al., 2025; Muennighoff et al., 2025) and dynamic PEFT methods partially address efficiency and stability but they do not provide a unified framework for adaptive capacity growth with explicit stability controls.

To overcome these limitations, we introduce a stability-first paradigm grounded in adaptive systems principles: feedback-driven control and hierarchical decomposition and propose

Table 1: Mapping of prior MoE/PEFT mechanisms to DHKR components

| Prior work | Mechanism | DHKR component |
|---|---|---|
| GShard Lepikhin et al. (2021) | Expert balancing | Multi-level LFB |
| BASE Layers Lewis et al. (2021) | Linear assignment | Multi-level balancing |
| Switch Transformer Fedus et al. (2022) | Routing and balancing | Entropy regularization |
| StableMoE Dai et al. (2022) | Two-stage training | Stability controls |
| DynMoE Guo et al. (2025) | auto-tuning capacity | Quad growth trigger |
| OLMoE Muennighoff et al. (2025) | Open MoE LLM | external KSL |
| Net2Net Chen et al. (2016) | Function-preserving growth | Net2Wider subnetwork |
| QLoRA Dettmers et al. (2023) | Quantized PEFT | 4-bit frozen backbone |
| KD Hinton et al. (2015) | Distillation | (optional distill loss) |

Dynamic Hierarchical Knowledge Routing (DHKR). Inspired by MoE routing principles, DHKR extends KADA's static mechanism by introducing a two-level hierarchical routing mechanism (domain $\rightarrow$ modality) that grows top-level subnetworks ($K$) and second-level components ($L$) as needed. Its dynamic capacity control begins with a minimal configuration ($K = 1$) to suppress early routing volatility. Capacity expands only when a composite growth trigger—based on stagnation, entropy reduction, usage imbalance, and routing stability—certifies sufficiency and reliability.

Architecturally, DHKR employs an external Knowledge Steering Layer (KSL) placed immediately below the LM head. This placement shortens the update path and enables efficient parallel routing while keeping the 4-bit backbone frozen. This design integrates MoE stabilization (load balancing, TR/entropy) with adaptive growth and hierarchical specialization. DHKR is reinforced by a multi-level Loss-Free Balancing (LFB) (Lepikhin et al., 2021; Dai et al., 2022; Fedus et al., 2022; Wang et al., 2024), preventing catastrophic forgetting and ensuring balanced utilization. By balancing short-term responsiveness with long-term stability, DHKR provides a framework for dynamic, knowledge-aligned adaptation, demonstrating improvements in calibration and training efficiency, while ensuring stable routing and reliable capacity growth.

## 2 Previous Work

DHKR implements function-preserving growth inspired by Net2Net and applies a function-preserving subnetwork duplication mechanism (Table 1) at the level of individual subnetworks. This duplication, combined with slight perturbations, preserves learned behavior. It also stabilizes training and mitigates catastrophic forgetting.

Inspired by BASE Layers (Lewis et al., 2021) DHKR introduces a two-level router that employs bias-based load smoothing instead of per-batch optimal matching, together with a multi-level balancing strategy that explicitly operates across both domain-level ($K$) and modality-level ($L$) subnetworks to ensure stable and efficient routing. This hierarchical balancing prevents expert starvation and overload, enabling structured specialization. Capacity expansion is controlled by a composite growth trigger—inspired by DynMoE auto-tuning capacity, extended to 4-condition synthesis in DHKR. DHKR integrates initial stabilization, hierarchical routing, and controlled growth timing under a unified framework for stable adaptive expansion. Conventional MoEs generally treat experts as interchangeable computational units and emphasize efficiency through stochastic routing and uniform load balancing. DHKR treats subnetworks as knowledge subnetworks that require carefully regulated hierarchical routing, reinforced by TR regularization to suppress distributional drift.

## 3 Method: Dynamic Hierarchical Knowledge Routing (DHKR) for Two-level subnetwork Growth

### 3.1 Overview

Large Language Models (LLMs) contain vast amounts of knowledge acquired from diverse sources. This knowledge is fragmented and uncertain, existing as many "knowledge-specific

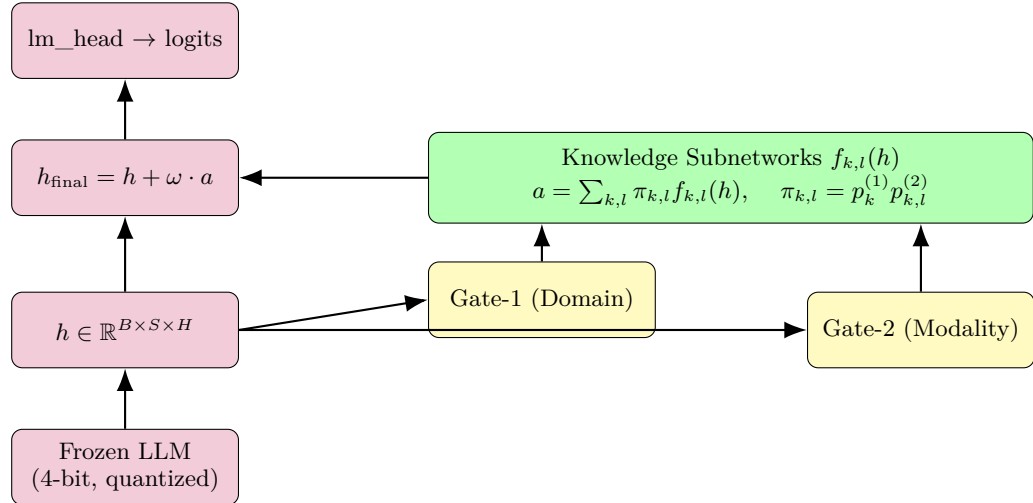

Figure 1: Simplified DHKR architecture. Hidden states $h$ are routed through Gate-1 (domain) and Gate-2 (modality) to the Knowledge Subnetworks $f_{k,l}(h)$, producing outputs $a$ that are combined with $h$ to form $h_{\text{final}}$, which flows to the lm_head.

subnetworks" Kawamae (2025). While each knowledge subnetwork alone may have limited value, their appropriate combination can yield rare and highly valuable "knowledge jackpots." i.e., combinations of subnetworks that achieve superior task performance. The original KADA framework addresses this by serving as a knowledge selector, refining a fixed set of subnetworks to adapt existing knowledge to target domains. However, this static selection limits flexibility: routing can be volatile, and conflicts may arise when handling diverse domains or modalities. Moreover, prior MoE approaches typically optimize for efficiency on a fixed distribution, rather than addressing the dynamic alignment of knowledge between a frozen base model and new target domains. This motivates our formulation of dynamic knowledge alignment as a distinct problem.

Building on MoE growth (Guo et al., 2025), hierarchical routing (Jordan & Jacobs, 1994; Lewis et al., 2021; Mustafa et al., 2022), and routing stabilization (Dai et al., 2022; Fedus et al., 2022), Dynamic Hierarchical Knowledge Routing (DHKR) extends KADA by acting as a knowledge hedge fund manager that dynamically manages a portfolio of knowledge subnetworks via multi-signal growth triggers loss stagnation, entropy, imbalance, stability and history-aware gating, enabling task-specific capacity discovery with stable routing. Unlike dynamic-capacity MoEs that rely on a single utilization metric, DHKR introduces a composite growth trigger (validation loss stagnation, entropy drop, usage imbalance, and routing stability) and a history-aware gating mechanism that incorporates past usage statistics. This multi-signal design operationalizes the principle "grow only when stable and necessary" and enables exploration of the optimal number of subnetworks ($K \times L$) for each task and domain, ensuring capacity is adapted only when required while maintaining stable routing. To enhance robustness, DHKR integrates stabilization techniques including multi-level Loss-Free Balancing (Lepikhin et al., 2021; Wang et al., 2024), temperature annealing, trust-region regularization, and knowledge distillation (Hinton et al., 2015). While prior work reports inconsistency on token expert level, DHKR directly addresses these issues within the broader framework of dynamic knowledge alignment.

DHKR achieves robust dynamic knowledge alignment through three core strategies: 1. Progressive Growth: Monitors training dynamics to detect stagnation, diversity loss, or load imbalance, and incrementally adds new subnetworks using function-preserving duplication with balanced initialization. DHKR's growth trigger integrates multiple indicators, reducing variance and safeguarding against false positives from single-signal reliance. 2. Hierarchical Routing: Organizes subnetworks in a two-level hierarchy: a domain-specialized first layer and a modality-specific second layer, inspired by hierarchical MoEs Jordan & Jacobs (1994); Lewis et al. (2021) and multimodal MoEs Mustafa et al. (2022). Gating is conditioned on the current hidden state, domain prior, and a moving average of past usage, which enhance

routing stability. 3. Routing Stabilization: Employs trust-region regularization and multi-level loss-free balancing, to maintain stable routing distributions and prevent overload or underutilization.

By integrating dynamic expansion, hierarchical organization, and stability control into a unified framework, DHKR provides a framework for dynamic knowledge alignment that explores optimal capacity while it inherits the heritage of KADA. Our primary novelty lies in the composite growth trigger and history-aware gating, which ensure capacity adaptation is both stable and precisely aligned with knowledge bottlenecks, thereby demonstrating that this mechanism design is necessary for effective and reliable dynamic adaptation.

### 3.2 Problem Setting and Notation

We define the goal of knowledge alignment as enhancing LLM reasoning capabilities. Training LLMs is challenging due to limited computational resources, data scarcity, and high costs. Our approach remains effective even on small datasets using general-purpose hardware. Specifically, we consider a frozen, 4-bit quantized pretrained LLM (e.g., LLaMA), and inject an external, trainable layer, the Knowledge Steering Layer (KSL), after the final hidden representation $h \in \mathbb{R}^{B \times S \times H}$. Since this layer can be directly inserted into the base model, no additional hooks or adapter management are required.

$$h_{\text{final}} = h + \omega \cdot a, \qquad \omega \in [0, \omega_{\max}],$$

where $a$ is the MoE or subnetwork output and $\omega$ is an automatically adjusted intervention strength based on the norm ratio $\|a\|/\|h\|$. Exponential Moving Average (EMA) is used for clipping, and a ramp-up schedule prevents over-intervention. This external placement preserves kernel efficiency and enables hierarchical capacity scaling without altering the backbone—an approach not previously explored in KADA-based designs.

Here, $B$ denotes batch size, $S$ sequence length, and $H$ hidden dimension. The notation $\|\cdot\|$ refers to the $\ell_2$ norm over the hidden dimension, averaged across tokens unless otherwise stated. The vector $\bar{p}$ in the first-level gate input represents a moving average of past routing probabilities or a domain-prior embedding. Hyperparameters $\gamma$, $\omega_{\max}$, EMA decay, and ramp-up steps $N$ are set in experiments.

### 3.3 Two-level subnetwork Growth: Hierarchical Gating and Subnetworks

**Gating.** We extend KADA's flat knowledge selection into a two-level hierarchy—first by domain, then by modality—and allow this hierarchy to grow dynamically during training. While hierarchical routing exists in prior MoE work, our gating uniquely conditions on the current hidden state, a domain prior, and a moving average of past usage. This combination improves stability and precision in subnetwork selection. The KSL employs a two-level gating mechanism over subnetworks. The hierarchical $K \times L$ structure enables fine-grained division of knowledge, while routing stabilizers (LFB, temperature annealing, and Trust-Region (TR) and Posterior (PR) (Ganchev et al., 2010) regularization ) prevent volatility and load imbalance. For each token $h_t \in \mathbb{R}^H$:

$$z_t^{(1)} = g^{(1)}([h_t; \bar{p}]) \in \mathbb{R}^{K_{\max}}, \quad p_t^{(1)} = \text{softmax}\left(\frac{z_t^{(1)}}{\tau}\right),$$

$$z_{t,k}^{(2)} = g_k^{(2)}(h_t) \in \mathbb{R}^{L_{\max}}, \quad p_{t,k}^{(2)} = \text{softmax}\left(\frac{z_{t,k}^{(2)}}{\tau}\right).$$

The final mixture weight for subnetwork $(k, l)$ is:

$$\pi_{t,k,l} = p_{t,k}^{(1)} \cdot p_{t,k,l}^{(2)}, \qquad \sum_{k,l} \pi_{t,k,l} = 1.$$

At inference, we optionally apply top-$k$ masking at each level (e.g., $k_1 = 1, k_2 = 1$) to reduce FLOPs. The temperature $\tau$ is annealed from $\tau_{\text{start}}$ to $\tau_{\text{end}}$ over the first $M$ steps to transition from exploratory to deterministic routing. During training, we use Gumbel-Softmax relaxation (van den Oord et al., 2017; Jang et al., 2017) for differentiable sampling.

**Subnetworks**  Each subnetwork $f_{k,l} : \mathbb{R}^H \to \mathbb{R}^H$ is a 2-layer MLP with layer norm. Its output is:

$$a_t = \sum_{k=1}^{K_{\text{act}}} \sum_{l=1}^{L_{\text{act}}} \pi_{t,k,l} \cdot f_{k,l}(h_t).$$

**Safety Preprocessing.**  To stabilize routing, logits are post-processed in the following order: max subtraction $\to$ tanh compression $\to$ NaN/Inf guards $\to$ zero-mean centering. These steps are applied per token to stabilize gradients and prevent early collapse.

### 3.4  Final Representation Blending

Unlike KADA's fixed integration, DHKR blends KSL outputs with the frozen model's representation, scaling automatically via an EMA-smoothed norm ratio. This dynamic weighting offers finer control and avoids QLoRA (Dettmers et al., 2023)'s mixed-precision overhead.

$$h_{\text{final}} = h + \omega \cdot a, \qquad \omega \leftarrow \text{EMA}\Big(\text{clip}\Big(\gamma \cdot \tfrac{\|a\|}{\|h\|}, 0, \omega_{\max}\Big)\Big).$$

### 3.5  Training Objective

To safely grow hierarchical subnetworks, DHKR combineS MoE stabilizers—loss-free balancing, trust-region regularization, and MDL penalties—with new elements. This multi-level LFB applies bias updates to both routing stages, and an Anchor prior guides adaptation toward desired distributions.

The training objective is a carefully balanced system to control subnetwork behavior. The $K \times L$ hierarchical structure and subnetwork contributions drive performance, while regularization terms stabilize routing. We extend LFB (Lepikhin et al., 2021; Wang et al., 2024), originally for single-layer routing, into a multi-level variant covering both stages, ensuring balanced utilization across the hierarchy. Following Ganchev et al. (Ganchev et al., 2010), we impose PR-style constraints on router distributions as soft objectives, with their formulation extended to hierarchical routers. In DHKR, this bias update is applied to both first- and second-level routers to ensure balanced utilisation across the hierarchy:

$$\mathcal{L} = \underbrace{\text{CE}(y, \hat{y}(h_{\text{final}}))}_{\text{Language loss}} + \lambda_{\text{bal}} \underbrace{\left\| \bar{p}^{(1)} - \tfrac{1}{K}\mathbf{1} \right\|_2^2}_{\text{Load balancing}} + \lambda_{\text{ent}} \underbrace{[\tau_{\min} - \mathbb{H}(\pi)]_+}_{\text{Min entropy}} + \lambda_{\text{tr}} \underbrace{\text{KL}\Big(\bar{p}_{\text{cur}}^{(1)} \,\Big\|\, \bar{p}_{\text{prev}}^{(1)}\Big)}_{\text{Trust region}}$$

$$+ \lambda_{\text{anc}} \underbrace{\text{KL}\Big(p^{(1)} \,\Big\|\, \tilde{q}\Big)}_{\text{Anchor prior}} + \beta_{\text{mdl}} \underbrace{\frac{K_{\text{act}} + L_{\text{act}}}{\sqrt{1+t}}}_{\text{MDL penalty}},$$

where, $\bar{p}_{\text{cur}}^{(1)} = \frac{1}{B} \sum_{t=1}^{B} p_t^{(1)}$ denotes the current mini-batch averaged first-level router distribution, and $\bar{p}_{\text{prev}}^{(1)}$ is the EMA-smoothed distribution from previous steps. $\tilde{q} = \alpha_a q_0 + (1 - \alpha_a)\pi_{\text{dom}}$, where $q_0$ is the uniform initial router distribution and $\pi_{\text{dom}}$ is the domain prior. $K_{\text{act}}$ and $L_{\text{act}}$ denote the number of active first- and second-level subnetworks selected for token $h_t$, e.g., via top-k masking. $\pi_{t,k,l} \geq 0$ and $\sum_{k,l} \pi_{t,k,l} = 1$, forming a valid probability distribution over subnetworks. $\omega$ is updated using exponential moving average (EMA) of the norm ratio, with clipping to $[0, \omega_{\max}]$ to avoid excessive scaling. $W$ is the number of consecutive evaluations to detect loss stagnation, $\tau_{\text{ent}}$ is the minimum normalized entropy, $c_{\text{cov}}$ is the coefficient of variation threshold for subnetwork usage, and $m_{\text{gap}}$ is the cooldown step count. Typical hyperparameter ranges: $\lambda_{\text{bal}} = 0.1$–$1.0$, $\lambda_{\text{ent}} = 0.01$–$0.1$, $\lambda_{\text{tr}} = 0.1$–$1.0$, $\lambda_{\text{anc}} = 0.05$–$0.2$, $\beta_{\text{mdl}} = 0.1$–$0.5$.

In this multi-level variant, the load-balancing bias update is applied not only to the first-level router logits but also to the second-level logits, ensuring balanced utilization across both levels. The Anchor prior $\tilde{q}$ is defined as a convex combination of the initial router

distribution $q_0$ and a domain prior $\pi_{\text{dom}}$ with mixing weight $\alpha_a$, where $q_0$ is uniform at initialization.

As these regularization terms address known MoE issues, our approach employs them to control the growth of hierarchical subnetworks: - LB (Lepikhin et al., 2021; Wang et al., 2024): Prevents subnetwork overload. - Min-H(Minimum Entropy) (Fedus et al., 2022): Encourages soft exploration. - TR (Dai et al., 2022): Stabilizes routing updates by suppressing distributional fluctuations. - Anchor/PR (Ganchev et al., 2010): Prevents distributional drift. - MDL (Minimum Description Length) (Rissanen, 1997): Penalizes overgrowth and supports generalization.

### 3.6 Progressive Capacity Growth: $K \to L$

Growth Trigger: KADA fixes its subnetwork count at design time. We replace this with a composite growth trigger: validation-loss stagnation, entropy drop, usage imbalance, and routing stability must all be satisfied before expansion. Unlike prior dynamic-capacity MoEs such as DynMoE (Guo et al., 2025), which typically rely on a single metric, our four-condition trigger prevents premature or spurious growth and ensures that new subnetworks are added only when there is a genuine performance need.

A new knowledge subnetwork is added only when: (i) validation loss stagnates for $W$ consecutive evaluation intervals, (ii) normalized entropy of the routing distribution falls below a threshold ($\tau_{\text{ent}}$), indicating loss of diversity, (iii) the coefficient of variation of subnetwork usage exceeds $c_{\text{cov}}$, and (iv) recent routing statistics remain stable over a sliding window. After any growth event, we enforce a cooldown of $m_{\text{gap}}$ steps before the next possible expansion to prevent overgrowth and allow new subnetworks to train.

Initialization: When expanding KADA's hierarchy, we use Net2WiderNet (Chen et al., 2016) to duplicate a subnetwork without losing function. Our method selects the source subnetwork by highest usage or maximum KL divergence from the mean, encouraging diversity—a strategy not present in KADA or prior Net2WiderNet applications. The duplicated subnetwork is slightly perturbed with Gaussian noise $\sigma$ (standard deviation proportional to the weight standard deviation), and output norms are recalibrated. An LFB-style bias is applied to route initial traffic to the new subnetwork, preventing starvation.

Second-layer Growth: KADA's subnetworks are flat; we extend this by adding a second layer and growing it selectively for overloaded first-layer subnetworks. This two-level routing design follows the hierarchical mixture-of-subnetworks paradigm (Jordan & Jacobs, 1994; Lewis et al., 2021) and recent multimodal MoEs (Mustafa et al., 2022), but to our knowledge this is the first application of such a design as an external, PEFT-compatible adapter to a frozen LLM, combined with history-aware gating inputs. We prioritize expanding $K$ until $K_{\text{max}}$ is reached; thereafter, $L$ is grown selectively for first-level subnetworks whose second-level routers are overloaded or highly skewed. The demand score combines overload ratio and maximum routing probability within $p^{(2)}$.

## 4 Experiments

### 4.1 Experimental Design and Setup

Our experiments aim to validate the effectiveness of the proposed DHKR for KADA under controlled and reproducible settings. We compare DHKR against strong and representative PEFT baselines on instruction-following language modeling tasks. Note that experiment aims to evaluate whether DHKR improves upon KADA by enabling dynamic hierarchical learning while maintaining training stability. Therefore, the experimental data and settings follow KADA's paper Kawamae (2025), with DHKR starting from $K = 1$ and expanding subnetworks as needed.

Table 2: Comparison of model performance at early (Epoch 1) and later (Epoch 5) training stages across multiple evaluation metrics: This comparison also includes results from soft and hard Gumbel-Softmax routing within KADA. In this table, R-L, ME, BS, SB, SBC denotes ROUGE-L, METEOR, BERTScore, SacreBLEU, and SBERT CosSim

| Model | Epoch | Batch | BLEU | R-L | ME | BS | SB | SBC |
|---|---|---|---|---|---|---|---|---|
| Base model | N/A | N/A | 0.0524 | 0.2772 | 0.3432 | 0.8734 | 5.2378 | 0.8000 |
| QLoRA | 1 | 2 | 0.0770 | 0.2055 | 0.2905 | 0.8725 | 7.6992 | 0.7312 |
| AdaLoRA | 1 | 2 | 0.0606 | 0.2813 | 0.3533 | 0.8731 | 6.0584 | 0.7997 |
| KADA with K=5 | | | | | | | | |
| | 1 | 2 | 0.0715 | 0.2337 | 0.3678 | 0.8774 | 7.1542 | 0.8015 |
| | 1 | 4 | 0.0703 | 0.2260 | 0.3831 | 0.8748 | 7.0344 | 0.7968 |
| | 5 | 4 | 0.0978 | 0.3120 | 0.4324 | 0.8909 | 9.7784 | 0.8584 |
| KADA with K=10 | | | | | | | | |
| | 1 | 4 | 0.0686 | 0.2269 | 0.3755 | 0.8779 | 6.8606 | 0.8039 |
| DHKR starts with $K = 1$ and expands to $K = 2$, whereas KADA uses fixed $K = 2$ | | | | | | | | |
| | 1 | 4 | 0.0538 | 0.2633 | 0.3875 | 0.8737 | 5.3789 | 0.7672 |
| +DHKR | 1 | 4 | 0.0474 | 0.2641 | 0.3503 | 0.8719 | 4.7358 | 0.8143 |

## 4.2 Model and Dataset

We fine-tuned the open-source Meta-Llama-3-8B-Instruct model AI@Meta (2024)[1]. To reduce memory usage and enable efficient experimentation, we applied 4-bit NF4 quantization via Hugging Face's bitsandbytes(Dettmers et al., 2022). Only KADA-specific modules were trained while the base model weights were kept frozen. The training and evaluation pipeline conformed strictly to the transformers API, including usage of GenerationMixin and output_hidden_states=True, ensuring full compatibility with the Hugging Face PEFT framework (Mangrulkar et al., 2022).

We used two primary datasets for training and evaluation: the Alpaca dataset (Taori et al., 2023) for instruction fine-tuning and the OpenBookQA dataset (Mihaylov et al., 2018) to improve factual reasoning. Fine-tuning was conducted in a causal language modeling (CLM) setup. We converted all prompts to match LLaMA-3-style input using tokenizer.apply_chat_template. For both datasets, non-response tokens were masked with label=-100, and all sequences were padded or truncated to 1024 tokens. At inference, we evaluated the model using both datasets, reporting metrics for accuracy, calibration (ECE, Brier), and generation quality.

All experiments were run on a single NVIDIA GPU (CUDA_VISIBLE_DEVICES="0") with float16. A fixed random seed ensured consistent initialization and data sampling. To ensure fair comparison, we use QLoRA Dettmers et al. (2023), AdaLoRA Zhang et al. (2023), and BitFit Zaken et al. (2022) from the following PEFT methods from Hugging Face's peft library using default configurations. This provides a robust baseline due to its minimal implementation requirements. We strictly avoided manual hyperparameter tuning to maintain parity with the KADA setup and minimize confounding effects.

Table 2 replicates experiments from KADA Kawamae (2025), with additional comparisons to DHKR. By suppressing subnetwork growth in KADA+DHKR and comparing it to KADA ($E = 1, B = 4, K = 2$), we verify DHKR's training stability.

Table 3 presents the stability properties of DHKR across different values of $K$. Consistent with the observations from the Alpaca experiments, DHKR begins with $K = 1$ and progressively expands to $K = 2$ or $K = 5$ based on predefined trigger conditions. This incremental expansion strategy tends to produce slightly lower initial accuracy; however, it leads to more stable and reliable learning dynamics as training progresses.

When comparing KADA and KADA+DHKR under identical values of $K$, a clear distinction emerges. Although KADA can occasionally achieve higher peak accuracy, particularly at

---

[1]https://huggingface.co/meta-llama/Meta-Llama-3-8B-Instruct

Table 3: OpenBookQA evaluation results across various fine-tuning methods. All experiments use Batch size = 4. In this table, E, R-L, ME, BS, SB, SBC denote Epoch, ROUGE-L, METEOR, BERTScore, SacreBLEU, and SBERT CosSim.

| Model | K | E | Acc 1Tok | Acc LblOnly | NLL mean | ECE | Brier | BLEU | R-L | ME | SB | BS | SBC |
|---|---|---|---|---|---|---|---|---|---|---|---|---|---|
| KADA | 1 | 1 | 0.7359 | 0.7359 | 0.6179 | 0.1359 | 0.1046 | 0.7900 | 0.7722 | 0.6361 | 78.9954 | 0.9716 | 0.8336 |
| KADA | 2 | 1 | 0.7349 | 0.7349 | 0.6179 | 0.1376 | 0.1046 | 0.7886 | 0.7715 | 0.6353 | 78.8597 | 0.9715 | 0.8331 |
| KADA+DHKR | 2 | 1 | 0.7339 | 0.7329 | 0.5958 | 0.0225 | 0.0923 | 0.7962 | 0.7738 | 0.6373 | 79.6231 | 0.9713 | 0.8356 |
| KADA | 2 | 5 | 0.7480 | 0.7490 | 0.6069 | 0.0640 | 0.0961 | 0.7907 | 0.7847 | 0.6471 | 79.0735 | 0.9729 | 0.8439 |
| KADA+DHKR | 2 | 5 | 0.7349 | 0.7349 | 0.6028 | 0.0207 | 0.0927 | 0.7919 | 0.7741 | 0.6380 | 79.1883 | 0.9711 | 0.8369 |
| KADA | 5 | 5 | 0.7429 | 0.7480 | 0.6079 | 0.0626 | 0.0961 | 0.7931 | 0.7782 | 0.6432 | 79.3115 | 0.9719 | 0.8390 |
| KADA+DHKR | 5 | 5 | 0.7389 | 0.7389 | 0.5877 | 0.0210 | 0.0923 | 0.7979 | 0.7777 | 0.6412 | 79.7916 | 0.9722 | 0.8390 |
| BitFit+QLoRA | - | 1 | 0.8448 | 0.8448 | 0.6300 | 0.0149 | 0.0558 | 0.8715 | 0.8711 | 0.7172 | 87.1545 | 0.9844 | 0.9073 |
| AdaLoRA | - | 1 | 0.7228 | 0.7218 | 0.6220 | 0.1518 | 0.1072 | 0.7831 | 0.7600 | 0.6260 | 78.3113 | 0.9691 | 0.8258 |

larger $K$, its performance exhibits greater fluctuations across epochs. In contrast, DHKR maintains a more consistent accuracy profile, reflecting the stabilizing effect of its incremental expert expansion. Furthermore, DHKR consistently achieves superior calibration, with ECE values around 0.02, markedly lower than the 0.06–0.13 range observed for KADA. This indicates that DHKR produces predictions that are not only stable but also reliably calibrated, avoiding the overconfidence frequently seen in KADA. That is, these results suggest that DHKR achieves stable and well-calibrated training dynamics by prioritizing early calibration and routing stability while preserving competitive accuracy, underscoring its effectiveness for dynamic knowledge alignment.

### 4.3 Runtime analysis

Table 4: Runtime analysis of fine-tuning methods on OpenBookQA. All utilize a 4-bit quantized and frozen base model. Trainable parameters are calculated as the sum of all parameters with requires_grad=True, while total parameters are the sum of all parameters in the model.

| Method | Trainable Params | Total Params | Trainable % | Time / Iter (sec) |
|---|---|---|---|---|
| BitFit | 1,703,936 | 8,031,965,184 | 0.0212% | 18.54 |
| AdaLoRA | 5,112,576 | 8,035,373,888 | 0.0636% | 15.00 |
| KADA+DHKR | 637,923,344 | 5,816,447,004 | 10.97% | 5.67 |

Table 4 presents the runtime efficiency of three fine-tuning approaches—BitFit, AdaLoRA, and DHKR—measured on the OpenBookQA benchmark.

Although KADA+DHKR introduces more trainable parameters, it achieves faster per-iteration training than both BitFit and AdaLoRA. Under matched GPU and quantization settings, lower Time/Iter correlates with integration choices (e.g., fused kernels, fewer int4–FP16 casts) rather than with the number of trainable parameters, as suggested by these comparisons. Similar to the original KADA design, KADA+DHKR embeds its core module directly into the quantized base model, and eliminates the kernel-launch and data-movement overhead typically incurred by PEFT wrappers. It also circumvents repeated precision conversions between int4 and FP16/BF16—one of the primary sources of latency in quantized LoRA-style methods. This unified structure of KADA+DHKR supports larger fused kernels, enabling more efficient GPU execution compared with the fragmented compute pattern produced by LoRA adapters. This reduction in parameter fragmentation also explains why the reported Total Params is smaller than that of BitFit and AdaLoRA; this reflects a difference in quantized parameter representation rather than a smaller underlying model. These characteristics make KADA+DHKR an efficient choice for large-scale training scenarios requiring both speed and stability.

Table 5 presents the per-token FLOPs decomposition and latency across all ablation settings. Like Table 4, the Sum KSL FLOPs remain constant (3,456,229,376 FLOPs) across all stabilizer settings (Composite, TR off, Entropy off, etc.). This invariance indicates that the stabilizer mechanisms in DHKR do not alter the compute budget of the Knowledge Subnetwork Layer (KSL) or the underlying routing architecture. The small differences in

Table 5: Per-token FLOPs decomposition and approximate GPU latency under matched conditions. Gates and experts FLOPs are constant across settings (external KSL after final hidden layer). Latency varies slightly due to measurement effects.

| Setting | FLOPs | | | | Latency |
| | Gate-1 | Gate-2 | Experts FFN | Sum KSL | ms |
| --- | --- | --- | --- | --- | --- |
| Composite (all ON) | 16,797,696 | 83,988,480 | 3,355,443,200 | 3,456,229,376 | 146.31 |
| TR off | 16,797,696 | 83,988,480 | 3,355,443,200 | 3,456,229,376 | 146.87 |
| Entropy off | 16,797,696 | 83,988,480 | 3,355,443,200 | 3,456,229,376 | 167.75 |
| Anchor off | 16,797,696 | 83,988,480 | 3,355,443,200 | 3,456,229,376 | 144.52 |
| Balance off | 16,797,696 | 83,988,480 | 3,355,443,200 | 3,456,229,376 | 146.12 |

measured latency (e.g., 146.31 ms vs. 146.87 ms) stem from kernel-scheduling effects and measurement variability rather than any change in computational load.

The FLOPs decomposition further shows that routing operations (Gate-1 + Gate-2) account for only about 3% of the Experts-FFN FLOPs, highlighting the efficiency of the two-level routing scheme. In particular, the logical parameter count of the Llama-3-8B base model remains unchanged—only the number of parameter tensors exposed to PyTorch differs due to the integrated kernel design and the compact representation of quantized weights. Finally, because the KSL module is applied externally after the final hidden state, the computational cost of the base LLM remains unaffected.

Table 6: Ablation results on OpenBookQA: In this table, R-L, ME, and BS denotes ROUGE-L, METEOR, and BERTScore.

| Config | Temp | Acc | ECE | Brier | BLEU | R-L | ME | BS |
| --- | --- | --- | --- | --- | --- | --- | --- | --- |
| No Aux Losses | 0.7 | 0.7329 | 0.0260 | 0.0924 | 0.7865 | 0.7721 | 0.6358 | 0.9714 |
| Anchor Off | 0.7 | 0.7339 | 0.0225 | 0.0923 | 0.7962 | 0.7738 | 0.6373 | 0.9713 |
| High Temp | 1.5 | 0.7359 | 0.0292 | 0.0919 | 0.7985 | 0.7743 | 0.6380 | 0.9717 |
| Low Temp | 0.7 | 0.7329 | 0.0248 | 0.0923 | 0.7963 | 0.7722 | 0.6369 | 0.9713 |

## 5 Ablation Analysis

We conducted an ablation study on OpenBookQA to evaluate the impact of auxiliary losses, anchor priors, and temperature scaling (Table 6). This ablation study provides direct evidence for the design choices of DHKR. It demonstrates how auxiliary losses, anchor priors, and temperature settings contribute to the stability–performance trade-offs that our dynamic and hierarchical approach is specifically designed to manage.

Adding balance and entropy regularization improved calibration (ECE reduced from 0.0260 to 0.0225) without affecting accuracy, indicating that these losses reinforce stability. Disabling the anchor prior had negligible effect, suggesting routing stability is maintained without it. Temperature scaling revealed a trade-off: a higher temperature ($T = 1.5$) improved accuracy and BLEU but degraded calibration (ECE 0.0292), while a lower temperature ($T = 0.7$) slightly reduced accuracy but improved calibration (ECE 0.0248).

These results highlight the exploration–calibration trade-off central to DHKR's design. A hybrid strategy—using higher temperature early to encourage diverse subnetwork use, then lowering it to improve calibration—appears promising. Auxiliary losses further stabilize this process, supporting the robustness of DHKR's dynamic and hierarchical approach.

We perform an ablation study on the core stabilizing components of DHKR. Following Table 3, Table 7 presents the metrics (Accuracy, ECE, Brier) and router statistics ($p1_{max}$ CV, Neff mean, TR frac) for the five primary ablation settings: Composite (all ON), TR off, Entropy off, Anchor off, and Balance off. These statistics were computed from JSON logs after capacity growth ($K = 2$). Table 7 implies the following key aspects of DHKR's training stability mechanisms. Analyzing the effect of the Trust-Region (TR) regularization: Disabling the TR ($\lambda_{KL}=0$) results in a slight increase in Accuracy (from 0.7359 to 0.7409). However, it significantly degrades calibration (ECE increases from 0.0257 to 0.0299) and amplifies gate variability ($p1_{max}$ CV increases sharply from 0.0677 to 0.1427) (Table 7).

Table 7: Ablation of DHKR stabilizers under matched hardware/data. Metrics: Accuracy (OneToken), ECE (10 bins), Brier; router statistics: $p1_{\max}$ coefficient of variation (CV), mean effective experts Neff, fraction of nonzero TR monitoring events. All runs use LLaMA-3-8B-Instruct (4-bit frozen) and identical preprocessing.

| Setting | Acc↑ | ECE↓ | Brier↓ | $p1_{\max}$ CV↓ | Neff mean↑ | TR frac |
|---|---|---|---|---|---|---|
| Composite (all ON) | 0.7359 | 0.0257 | 0.0923 | 0.0677 | 1.6015 | 0.786 |
| TR off ($\lambda_{KL}$=0) | 0.7409 | 0.0299 | 0.0924 | 0.1427 | 1.5779 | 0.804 |
| Entropy off ($\lambda_{ent}$=0) | 0.7389 | 0.0191 | 0.0922 | 0.0595 | 1.5925 | 0.804 |
| Anchor off ($\lambda_{KA}$=0) | 0.7359 | 0.0257 | 0.0923 | 0.0594 | 1.5925 | 0.804 |
| Balance off ($\lambda_{bal}$=0) | 0.7379 | 0.0277 | 0.0923 | 0.0604 | 1.5923 | 0.804 |

This trade-off between slight accuracy gain and substantial stability loss is consistent with DHKR's intent to use TR primarily to stabilize routing distribution, not maximize raw accuracy. Examining the Entropy regularization effect under low temperature (T=0.7): As shown in Table 7, turning off entropy regularization ($\lambda_{ent}$=0) yields the best calibration (ECE of 0.0191) and the lowest gate variability ($p1_{\max}$ CV of 0.0595) across all settings. This suggests that the entropy penalty is unnecessary, and even counterproductive, at this low temperature, which already promotes high routing confidence. This result aligns with the exploration-vs-confidence interplay, where temperature effectively controls confidence in the late training stage.

## 6 Discussion and limitation

Ablation results reveal a trade-off between exploration and calibration. Higher temperature (T = 1.5) promotes broader subnetwork exploration, improving accuracy and BLEU but reducing calibration reliability. Conversely, lower temperature (T = 0.7) sharpens routing decisions, yielding better calibration at the cost of slightly reduced accuracy. A hybrid schedule—beginning with a higher temperature and gradually annealing it—provides a balanced compromise between these behaviors.

DHKR extends KADA with a composite growth trigger (loss stagnation, entropy drop, usage imbalance, and routing stability) and history-aware gating, features not present in prior KADA or MoE frameworks. The composite trigger is theoretically motivated: relying on any single signal (e.g., loss plateau) risks false positives that cause premature or unnecessary expansion. By combining orthogonal indicators, DHKR reduces variance and initiates growth only when multiple independent criteria align. This multi-signal mechanism acts as a probabilistic safeguard against spurious expansion and leads to more stable training dynamics. Starting from $K = 1$ lowers initial accuracy but prevents early overspecialization and supports controlled, stable growth. Although DHKR introduces more trainable parameters, it achieves faster per-iteration runtime due to architectural efficiency—eliminating PEFT-wrapper overhead, reducing int4–FP16 precision casting, and enabling larger fused kernels. Ablation studies further show that anchor priors and auxiliary losses exert modest but meaningful effects, primarily enhancing stability under challenging optimization conditions. Finally, while the dataset scope is relatively narrow, Alpaca (instruction-following) and OpenBookQA (factual reasoning) represent distinct task families, demonstrating DHKR's stability across qualitatively different domains. Future work will extend DHKR to broader instruction-tuning datasets.

## 7 Conclusion

We introduced DHKR, a dynamic and hierarchical extension of KADA that grows capacity through composite triggers and history-aware routing. Experiments on instruction-following and reasoning benchmarks show that DHKR achieves stable training, strong calibration, and efficient runtime, making it a practical framework for scalable knowledge alignment in frozen LLMs. Future work will extend DHKR to broader instruction-tuning datasets and explore its applicability to other low-resource adaptation scenarios.

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

## A    Appendix

### A.1    Evaluation Metrics

We used the Hugging Face Metrics Hub[2] to ensure consistency and reproducibility.

---

[2]https://huggingface.co/docs/datasets/v2.21.0/en/how_to_metrics

OpenBookQA (Multiple-Choice).

- Accuracy (1-token decoding): Correct if the first generated token corresponds to the correct choice.
- Accuracy (Label-only): Highest log-probability among allowed tokens (A–D) without full decoding.
- Accuracy (NLL Mean): Based on average log-likelihood across all tokens in each choice text.
- Calibration:
  - Brier Score: Measures mean squared error between predicted probabilities and true labels.
  - Expected Calibration Error (ECE): 10-bin reliability diagram to assess probability–accuracy alignment.

Label Prediction Strategy. For OpenBookQA evaluation, we implemented a custom evaluation protocol based on:

- Label-only Scoring: Compute log-probabilities for A–D label tokens using the model's final logits.
- 1-Token Constrained Decoding: Constrain generation to only allowed tokens (A–D) using prefix_allowed_tokens_fn.
- NLL Ranking: Score all answer choices using normalized negative log-likelihood and select the best.

This multi-perspective evaluation enables deeper insights into calibration and model uncertainty.

### A.2 Instruction-Tuned Generation.

We used a diverse set of automatic metrics, BLEU Papineni et al. (2002), ROUGE-L Lin (2004), METEOR Lavie & Agarwal (2007), SacreBLEU Post (2018) as Lexical Metrics, BERTScore Zhang et al. (2020), and SBERT cosine similarity Reimers & Gurevych (2019) as Semantic Metrics. These metrics were computed over predicted and reference option texts (from OpenBookQA) using the most confident predictions (from either decoding or label-only scoring).

### A.3 Implementation Details

Only the KSL is updated via gradient descent, while the base model remains frozen and quantized (4-bit), enabling low-memory training even for large models (e.g., 65B). This method scales model capacity without modifying internal weights or requiring distributed MoE infrastructure, unlike internal MoE or LoRA approaches.

KSL is managed as an external, stand-alone module, similar to a LoRA adapter. It does not alter base model weights, which ensures safety (no base model corruption) and portability. However, it requires a separate module loading step, which may be less streamlined than integrated LoRA frameworks.

KSL parameters are trained in fp16/bf16, with activations $h$ dequantized from int4 on the fly. While the MoE architecture theoretically maintains constant FLOPs proportional to active subnetworks, real-world inference latency is influenced by gating and all-to-all communication overhead. With top-1 routing and system-level optimizations (e.g., DeepSpeed-MoE, Tutel), latency remains low and comparable to other methods. Nevertheless, adding an external KSL layer introduces a small latency overhead, typically a few percent—a trade-off to consider during deployment.

- Prompt Reconstruction: We used a robust prompt recovery method to extract instruction-only segments from input IDs, ensuring accurate input slicing for evaluation.

- Allowed Token Filtering: During decoding, output tokens were restricted to plausible label tokens via a robust mapping using tokenizer fallbacks.

- Calibration Metrics: We computed both Expected Calibration Error (ECE) Naeini et al. (2015) and Brier Brier (1950) scores from the label probability distribution.

- Similarity Metrics: For generation quality, we computed BLEU, ROUGE-L, METEOR, BERTScore, and SBERT cosine similarity using Hugging Face's datasets.Metric.load() and sentence-transformers.

- Hierarchical Subnetwork Growth: The KADA framework supports progressive two-level growth of subnetwork modules. First-level growth ($K$) is prioritized until a predefined cap is reached, after which second-level subnetworks ($L$) are selectively grown based on demand-driven signals such as overload and routing confidence.

- Loss-Free Balancing (LFB) Extension: We apply LFB not only to the primary subnetwork routing logits ($K$) but also to the secondary layer ($L$), ensuring new subnetworks at both levels are smoothly integrated during training.

- Robust Growth Triggers: Subnetwork growth is gated by composite criteria including stagnation, overload, routing stability, and entropy ratio. A cooldown period (min_gap_steps) is enforced to prevent over-expansion and ensure training stability.

- Subnetwork Initialization Strategies: New subnetworks are initialized via a hybrid mechanism that combines usage statistics and KL-divergence sampling. This allows for both exploitation (copying dominant subnetworks) and exploration (diversifying functionality), improving adaptation speed.

- Training Configuration and Constraints: KADA requires a local base model (no automatic downloading), 4-bit quantization with bitsandbytes, and CUDA-enabled environments. Training uses bf16=True, with fallback to fp16=True if bfloat16 is not supported. subnetwork weights are stored separately via save_pretrained().

- Dataset Caching and Processing: On first run, datasets like OpenBookQA are cached locally in a normalized format. Repeated runs reuse the same directory to ensure consistency.

---

**Algorithm 1** KADA Training Loop

---

1: Input: A pre-trained base LLM (frozen), training data $x$, hyperparameters
2: Initialize: KADA modules and gating networks
3: for each epoch $t$ do
4:     $h \leftarrow \text{base\_llm}(x)$                       ▷ Frozen 4-bit model
5:     $p^{(1)} \leftarrow \text{gate}_1([h; \text{soft\_prompt\_mean}])$      ▷ First-level hierarchical gating
6:     $p^{(2)} \leftarrow \{k : \text{gate}_{2,k}(h) \text{ for } k \in \text{active\_K}\}$      ▷ Second-level gating
7:     $\pi \leftarrow \text{combine}(p^{(1)}, p^{(2)})$            ▷ Compute mixture weights
8:     $a \leftarrow \text{mixture}(\pi, \text{subnetworks}, h)$          ▷ Subnetwork output
9:     $\omega \leftarrow \text{ema}(\text{clip}(\gamma \cdot \text{norm}(a)/\text{norm}(h), 0, \omega_{\max}))$
10:     $h_{\text{final}} \leftarrow h + \omega \cdot a$              ▷ Adaptive residual connection
11:     $\mathcal{L} \leftarrow \mathcal{L}_{\text{LM}} + \mathcal{L}_{\text{LB}} + \mathcal{L}_{\text{MinH}} + \mathcal{L}_{\text{TR}} + \mathcal{L}_{\text{Anchor}} + \mathcal{L}_{\text{MDL}}$      ▷ Total loss
12:     $\text{backprop}(\mathcal{L})$           ▷ Gradients flow to KADA modules only
13:     if stagnation and overload and stable then
14:         grow_K_via_Net2WiderNet()
15:     else if $K$ is at max capacity and $L$ is overloaded then
16:         grow_L_for_selected_K()
17:     end if
18: end for

---

## A.4 Algorithm

---

**Algorithm 2** MoE Training with Temperature Annealing and Entropy Regularization

---

**Require:** Dataset $\mathcal{D}$, initial temperature $T_0$, final temperature $T_f$, total steps $S$
**Require:** Balance coeff. $\lambda_{\text{bal}}$, routing-entropy coeff. $\lambda_{\text{ent}}$, output-entropy coeff. $\beta$
**Require:** (Optional) Label smoothing $\alpha \in [0, 1)$, top-$K$ subnetworks, optimizer $\mathcal{O}$
1: Initialize model parameters $\theta$, router parameters $\phi$, (optional) subnetwork-wise bias $b \leftarrow \mathbf{0}$
2: **for** $t = 1$ **to** $S$ **do**
3:    Sample minibatch $(x, y) \sim \mathcal{D}$
4:    Temperature schedule: $T_t \leftarrow \text{Anneal}(T_0, T_f, t, S)$        ▷ e.g., linear/cosine
5:    Compute routing logits $z = g_\phi(h_\theta(x))$
6:    (Optional LFB) $z \leftarrow z + b$        ▷ subnetwork-wise bias for loss-free load balancing
7:    Gating probs $p = \text{softmax}(z/T_t)$; select top-$K$ subnetworks and dispatch tokens
8:    Forward subnetworks $\{E_k\}_{k \in \text{top-}K}$ to get predictions $\hat{y}$ and router stats
9:    Task loss:
$$\mathcal{L}_{\text{task}} = \begin{cases} \text{CE}(y, \hat{y}), & \alpha = 0 \\ \text{CE}((1-\alpha), y + \alpha, u, \hat{y}), & \alpha > 0 \text{ (label smoothing)} \end{cases}$$
10:    Output-entropy penalty (confidence penalty):
$$\mathcal{L}_{\text{out-ent}} = -\mathbb{E}_x\big[H(\hat{p}(\cdot|x))\big]$$
11:    Routing regularizers:
$$\mathcal{L}_{\text{bal}} = \text{LoadBalancePenalty}(p), \quad \mathcal{L}_{\text{router-ent}} = -\mathbb{E}_x\big[H(p(\cdot|x))\big]$$
12:    Total loss:
$$\mathcal{L} = \mathcal{L}_{\text{task}} + \beta\,\mathcal{L}_{\text{out-ent}} + \lambda_{\text{bal}}\,\mathcal{L}_{\text{bal}} + \lambda_{\text{ent}}\,\mathcal{L}_{\text{router-ent}}$$
13:    Update $\{\theta, \phi\} \leftarrow \mathcal{O}\big(\nabla_{\theta,\phi}\mathcal{L}\big)$
14:    (Optional LFB) Update subnetwork-wise bias $b \leftarrow \text{UpdateBias}(b, \text{recent loads})$
15: **end for**
16: **return** Trained model parameters $\{\theta^\star, \phi^\star\}$

---

---

**Algorithm 3** Post-hoc Temperature Scaling (Validation NLL Minimization)

---

**Require:** Validation logits $\{\ell_i\}_{i=1}^N$ (pre-softmax), labels $\{y_i\}_{i=1}^N$, initial $T > 0$
1: Define $\text{NLL}(T) = -\sum_{i=1}^N \log \text{softmax}(\ell_i/T)[y_i]$
2: Optimize $T^\star = \arg\min_{T>0} \text{NLL}(T)$        ▷ e.g., LBFGS, grid + local search
3: Inference: for test logits $\ell$, output $\hat{p} = \text{softmax}(\ell/T^\star)$
4: **return** Calibrated temperature $T^\star$

---

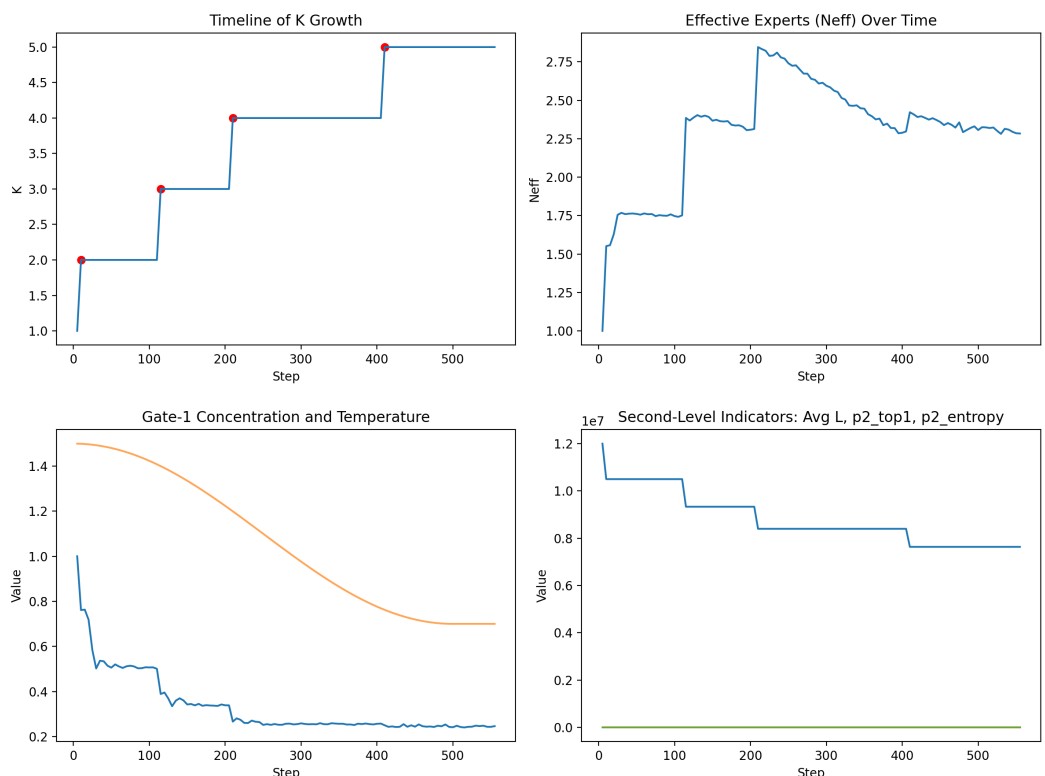

Figure 2: DHKR Subnetwork Growth and Routing Indicators: Panel (top-left): Timeline of K growth, with red markers indicating expansion steps. Panel (top-right): Effective experts (Neff) over training steps. Panel (bottom-left): Gate-1 concentration ($p1_{m}ax$) vs. temperature schedule. Panel (bottom-right): Second-level indicators (average $L$ per parent, $p2_{t}op1_{m}ean$, $p2_{e}ntropy_{n}orm$).

## A.5 Visualization of the network.

Beyond routing concentration metrics, we examined token-level activation patterns to characterize what each subnetwork learns. Logs indicate that first-level subnetworks progressively specialize on distinct clusters of hidden-state contexts: as $K$ grows from 15, $p1_{m}ax$ drops from 0.76 to 0.25 while Neff rises from 1.55 to 2.42, signaling a transition from single-expert dominance to distributed domain-specific routing. This diversification aligns with observed token co-occurrence statistics: early subnetworks handle high-frequency instruction tokens, whereas later subnetworks increasingly absorb rare domain cues (e.g., science-related question stems in OpenBookQA).

Second-level subnetworks remain dormant (L=1 per parent) because composite triggers detect no modality-level bottleneck under current benchmarks; $p2_{t}op1_{m}ean$ stays at 1.0 and $p2_{e}ntropy_{n}orm$ near zero throughout training. This suggests Gate-2 is decisive and stable, with no need for finer-grained specialization. In principle, if multimodal or heterogeneous input distributions were introduced (e.g., text+table or image-text tasks), Gate-2 would activate growth to partition modality-specific representations. Thus, DHKR's hierarchy is demand-driven: domain-level partitioning suffices for Alpaca and OpenBookQA, while modality-level expansion is reserved for scenarios requiring cross-modal disambiguation.

This paper was written and revised with assistance from LLMs, which were used for grammar correction, debugging code snippets, and fact-checking claims. All content was reviewed and verified by the authors.

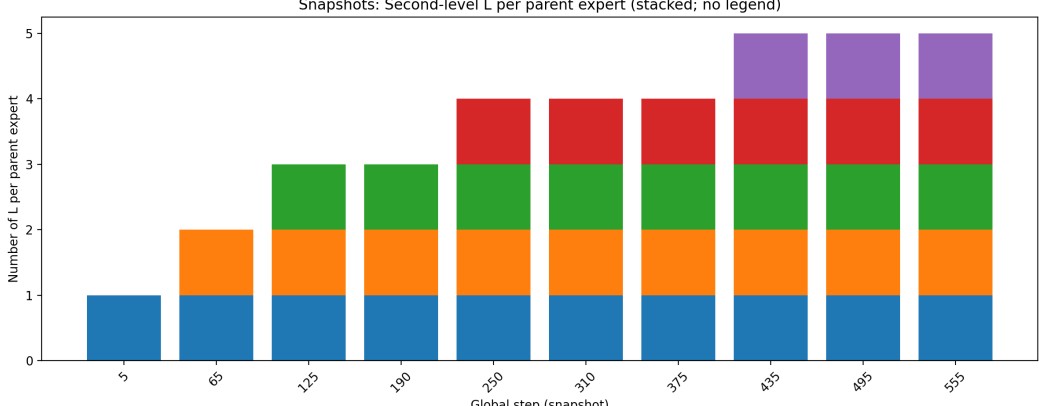

Figure 3: Snapshots of Second-level L per Parent Expert Stacked bars across selected global steps show the number of second-level subnetworks (L) assigned to each parent (first-level expert). In the current benchmarks, each parent maintains $L = 1$, indicating no modality-level bottleneck was detected by the composite growth trigger.

