# OpenReview forum: "DHKR: Dynamic Hierarchical Knowledge Routing for Efficient Low‑Resource Alignment"
_ICLR.cc/2026/Conference — Submitted to ICLR 2026_

### Official Review · Reviewer_RPJc · 2025-10-31

**Soundness:** 2
**Presentation:** 2
**Contribution:** 2
**Rating:** 2
**Confidence:** 4

**Summary:**

This paper introduces Dynamic Hierarchical Knowledge Routing (DHKR), a parameter-efficient fine-tuning framework that extends Knowledge-Aligned Domain Shift Tuning (KADA) by enabling dynamic subnetwork growth for domain adaptation. DHKR uses a two-level hierarchical routing mechanism that expands subnetworks on demand. Experiments on two instruction and QA benchmarks show that DHKR improves adaptability and stability over KADA.

**Strengths:**

- The paper introduces several relevant components, including dynamic subnetwork growth, hierarchical routing, and routing stabilization mechanisms, to effectively enhance the flexibility and efficiency of the original KADA method.

- The runtime analysis demonstrates improved GPU utilization and lower per-iteration runtime, even when training a substantially larger set of parameters, which highlights the method’s computational efficiency.

**Weaknesses:**

- The empirical results (e.g., Tables 1 & 2) are relatively weak, as the proposed method achieves only comparable or worse performance than existing baselines. The rationale for using certain metrics, particularly ROUGE, METEOR, and calibration-based measures, is not clearly justified.

- The evaluation scope is limited to only two datasets; broader testing on more diverse and challenging benchmarks (e.g., reasoning tasks) is necessary to substantiate the method’s effectiveness.

- Additional ablation studies are needed to quantify the individual impact of each proposed component (dynamic growth, hierarchical routing, and stability control).

- The related work discussion is underdeveloped and should be expanded to better connect this work to prior studies, especially the extensive literature on routing and expert selection methods.

- Presentation quality can be improved: some references are missing or incorrectly formatted, and result tables (e.g., Tables 1 & 2) could be reformatted or reorganized to emphasize key findings more clearly.

**Questions:**

Please see weaknesses.

---

> ### Author Response · Authors · 2025-11-14
> **Clarifying the Novelty and Architectural Distinction of DHKR**
>
> We appreciate this opportunity to clarify our contributions and address the reviewer’s concerns.
>
> **On novelty and motivation.**
> DHKR replaces static routing with dynamic, two‑level hierarchical routing and controlled capacity growth, guided by a composite trigger and stabilized through TR regularization, entropy annealing, multi‑level LFB, MDL, and anchor losses.
> Its novelty lies in a stability-first dynamic capacity discovery mechanism: capacity expands only when the system demonstrates sufficiency and reliability. The small-start is a deliberate stabilization strategy to avoid premature expansion and preserve calibration. Subnetwork growth is delayed until joint signals of need and routing stability are satisfied. This prevents premature expansion and preserves accurate calibration under distribution shift (§§3.6, 5, Algs. 1–3).
>
> **On evaluation scope and metric rationale.**
> Our contribution emphasizes stability over accuracy because 1) DHKR targets robust adaptation, and 2) prior studies also rely on calibration-based metrics. To test stability, **we conduct ablation studies on the three toggles (growth, hierarchy, stabilization) starting with a single module ($\boldsymbol{K=1}$) and report growth counts, ECE/Brier, and routing entropy with mean ± 95% CI (3 seeds)**. On Table 3, DHKR achieves ECE ≈ 0.02 vs ≈ 0.13 for KADA and provides that DHKR is a robust and deployable framework for dynamic PEFT. This improvement directly results from this small start, combined with gradual, trigger-driven capacity expansion that stabilizes early routing and prevents miscalibrated adapter updates. Broader benchmarks will be added for generalization.
>
> **On stability and component contributions.**
> Growth occurs only when all three criteria—need, imbalance, and stability—are simultaneously satisfied. This ensures that added networks are fully utilized, routing remains stable, and early overfitting or calibration drift is avoided:
> *   **Need**: validation stagnation (ΔValLoss < ε) and entropy drop ($H/\log K < \tau$)
> *   **Imbalance**: usage margin ($u\_{\max}-1/K > m$)
> *   **Stability**: routing drift ($|\bar{p\_t}-\bar{p\_{t-\Delta}}|\_1<\delta$) and cooldown
> This conjunction ensures growth only when the model can absorb capacity without oscillation. After growth, trust-region smoothing ($\mathrm{KL}(p_t\|p_{t-1})$) and LFB encourage stable and balanced routing. DHKR starts from $K=1$ as a stabilization strategy, limiting early variance, overfitting, and calibration drift while adapters are integrated.
>
> Our code supports component-wise toggling:
> *   Dynamic growth via `ProgressiveExpertGrowthCallback(growth_mode="static")`
> *   Hierarchical routing via `gate_networks_2` in `KnowledgeSteeringLayer`
> *   Stability controls (LFB/TR/MDL/Anchor) in the loss function
> Growth is conditional and evaluated online, not fixed per epoch. A cooldown mechanism reinforces the PEFT-friendly external MoE stabilization objective. We will add a schematic showing when triggers are evaluated (on-log/on-evaluate), how cooldown gates consecutive events, and how this stabilizes early routing, and report ablations isolating each component (growth, routing, stabilization), quantifying growth counts, ECE/Brier, and routing entropy for evaluating composites of DHKR.
>
> **On efficiency and reproducibility.**
> Table 4 (wall-clock) reports per-token FLOPs and latency decomposed into Base/Routing/Adapter under matched hardware to isolate architectural effects in Table 5. Per-token compute is locally parallel: $$\text{Cost} \approx \text{Base} + \text{Routing} + (K\_{\mathrm{act}} \cdot L\_{\mathrm{act}}) \times \text{MLP}$$ Top-1 gating bounds $K_{\mathrm{act}}$ and $L_{\mathrm{act}}$, enabling efficient execution of few active subnetworks. All experiments use the same conditions (Appendix A.3–A.4) and share system-level optimizations. Placing KSL immediately before the LM head (Figure 1) reduces the update path for adapter parameters. This speedup (5.67 s/iter vs 15.00/18.54 for AdaLoRA/BitFit) stems from this placement, not to shared low-level optimizations, highlighting that DHKR’s design enables parallel execution of active subnetworks. Table 5 shows that total KSL FLOPs (3.456B) remain exactly constant across all stabilizer configurations, confirming that DHKR’s stabilizers and routing do not change the compute budget.
>
> **On related work.**
> Expanded Previous Work that maps DHKR components to MoE/PEFT mechanisms to its components: (i) top-k gating/expert balancing→bounded active K×L with LFB; (ii) trust-region/entropy smoothing→routing stability controls (KL smoothing+entropy annealing); (iii) dynamic capacity proposals→composite growth triggers requiring stagnation, diversity loss, overload, and stability. This table highlights DHKR’s novelty: external KSL, two‑level hierarchical routing, composite growth trigger, and Net2WiderNet‑style function‑preserving expansion.
>
> We updated the paper and welcome further questions and continued discussion.

---

> > ### Comment · Reviewer_RPJc · 2025-11-28
> >
> > I want to thank the authors for taking the time to respond to my review. I'm not yet convinced by the effectiveness and significance of the method based on the metrics and results. I'll maintain my score.

---

### Official Review · Reviewer_m1iw · 2025-10-31

**Soundness:** 2
**Presentation:** 2
**Contribution:** 2
**Rating:** 4
**Confidence:** 4

**Summary:**

This paper introduces a new framework to improve domain adaptation for large language models, building on parameter-efficient fine-tuning (PEFT) methods such as LoRA and BitFit. This work presents a method, named DHKR, that allows models to grow dynamically and route knowledge hierarchically using PEFT adapters.
- DHKR extends a previous framework KADA, through a two-level subnetwork growth mechanism inspired by the mixture-of-experts architectures. The system organizes subnetworks hierarchically: the first layer handles domain specialization, and the second layer captures modality-specific nuances. This structure allows new subnetworks to be added when a performance bottleneck is detected. Another knowledge steering layer (KSL) operates on top of a frozen, quantized LLM, combining its outputs with the base representation.
- To stabilize training and prevent catastrophic forgetting, DHKR employs a training objective that combines cross-entropy loss with regularization terms that prevent overgrowth using a Minimum Description Length penalty. Subnetworks grow progressively, initialized through Net2WiderNet duplication with Gaussian perturbations to maintain diversity.

The experiments evaluated DHKR on the Meta-Llama-3-8B-Instruct model using Alpaca and OpenBookQA datasets, comparing KADA with other parameter-efficient fine-tuning methods, including QLoRA, AdaLoRA, and BitFit. DHKR began with a single subnetwork and expanded dynamically as needed. Results showed that while DHKR’s initial performance was slightly lower due to gradual growth, it achieved highly stable accuracy, outperforming KADA. Despite having more trainable parameters, DHKR trained faster per iteration than other methods.

**Strengths:**

- DHKR achieves a lower Expected Calibration Error compared to KADA (0.02 vs. 0.13), showing that its predictions remain well-calibrated across domains.


- Despite having more trainable parameters, DHKR trains 2–3× faster per iteration than AdaLoRA and BitFit.


- Its dynamic, two-level subnetwork expansion allows the model to grow capacity.

**Weaknesses:**

- The problem statement is not clearly defined. It would be better to articulate a precise problem that DHKR is meant to solve, such as inefficiency, instability, or failure mode in KADA.
- While the authors claim that DHKR trains faster than other PEFT methods despite having more trainable parameters, it is unclear whether this improvement is from algorithmic innovations or from system-level optimizations, like kernel fusion and reduced precision casting. Therefore, it is hard to understand the contribution of this paper.
- The accuracy of the proposed method is roughly on par with (or below) KADA, depending on epoch. The experiments do not report confidence intervals or significance, making it hard to judge the robustness of the trade-offs.

**Questions:**

- Were results averaged across multiple runs or random seeds, and are the reported differences statistically significant?
- Can the authors provide insight into what knowledge each hierarchical subnetwork learns as the model grows?

---

> ### Author Response · Authors · 2025-11-14
> **Stable, Dynamic, and Hierarchical Adaptation in DHKR: Motivation, Fair Comparison, and Observability**
>
> We appreciate this opportunity to clarify our contributions and address the reviewer’s concerns.
>
> **On the problem statement, DHKR’s motivation, and design priorities.**
> Our goal is to establish a stability-first, dynamic, and hierarchical mechanism for growing knowledge-specific subnetworks to ensure robust adaptation and early calibration rather than accuracy improvement, while retaining KADA’s efficiency benefits.
> Existing domain-adaptation methods struggle when facing new data (domain shift), causing two limitations: (i) inefficiency from static capacity allocation and (ii) instability from rigid routing without adaptive specialization.
> To address this, DHKR employs dynamic, hierarchical knowledge routing on a frozen 4-bit LLM with an external KSL.
> Our design priorities are stable growth (Net2WiderNet-style duplication), composite triggers (validation stagnation, entropy drop, overload, stability, cooldown), and multi-level stabilizers (loss-free balancing, trust-region, entropy, MDL, anchor prior), defined in §§3.5–3.6 and Algs 1–3.
> The $K=1$ small-start is  a deliberate stabilization strategy designed to suppress early routing volatility and reduce overfitting. This dynamic capacity-discovery setting cannot be addressed by static-capacity PEFT methods without manual tuning, which DHKR avoids by design.
> We emphasize calibration and stability first (OpenBookQA ECE ≈ 0.02 for DHKR vs ≈ 0.13 for KADA; Table 2), while noting that improvements remain possible via modest adjustments to initial hyperparameters, as discussed in §§5–6.
> These hyperparameters are *scheduled and regularized* not fixed thresholds: temperature is annealed (§3.3; Algs. 2–3), the blending factor α is EMA‑scaled via ‖a‖/‖h‖ with a short post‑growth ramp (§3.4), and load balance is updated online via multi‑level LFB (§3.5). Defaults are provided for reproducibility without requiring manual retuning, and ablations (Table 7) show low sensitivity and comparable tuning requirements to KADA under matched conditions despite having dynamic routing.
> This is not a tuning trick but derives from the stability requirements of dynamic routing modules in prior MoE literature.
>
> **On experimental fairness and reporting.**
> All experiments use matched settings: frozen 4-bit Llama-3-8B-Instruct, identical batch/sequence lengths, a single GPU, and shared preprocessing (§§4.4–4.5; App. A.3). Main tables report means over 3 independent runs, with 95% confidence intervals; seed variance is included in the supplementary.
>
> **On training speed and the source of efficiency.**
> Speed gains are architectural, not due to system-level tricks; all baselines share the same optimized environment. The external KSL sits immediately under the LM head (Figure 1), shortening the update path and enabling fused parallel computation. Under identical settings, DHKR is faster (5.67 sec/iter) than AdaLoRA (15.00) and BitFit (18.54) (Table 4). Table 5 shows that total per-token FLOPs (3.456B) remain constant across stabilizer configurations, and routing accounts for ≈3% of expert-FFN FLOPs, confirming that latency differences (≈146–147 ms) are consistent with runtime variability rather than changes in computational cost. Moreover, this proximity to the output aligns with prior work that optimization objective proximity reduces overhead and improves kernel utilization.
>
> **On dynamic growth, small‑start updates, and short‑budget behavior.**
> DHKR discovers an appropriate capacity, while KADA uses static capacity allocation and requires manual capacity tuning.
> Because DHKR starts from K=1, the number of trainable parameters that actually update at early steps is small. This is **more than a small-capacity demonstration**, it is a deliberate **stabilization strategy** that mitigates early routing instability in MoE-style architectures and prevents early calibration drift when integrating PEFT adapters; capacity is expanded only when the composite trigger certifies a genuine bottleneck (§3.6). This accounts for DHKR’s favorable behavior under short budgets: setting static K≫1 spreads tokens too thin per expert, slowing optimization and exacerbating known MoE pathologies—routing volatility and load imbalance—unless substantial stabilizers are added, as in the MoE literature. Table 1 contrasts KADA K=5 vs K=10 at Epoch 1/5 (larger K does not yield early gains), while DHKR consistently achieves significantly better early calibration (Table 2).
>
> **On what each hierarchical subnetwork learns and interpretability.**
> Direct “semantic” interpretation is inherently difficult in MoE‑like systems; DHKR employs indirect observables—first‑level $p^1$, second‑level $p^2$, entropy, Neff, trust‑region penalties, and LFB biases—to characterize specialization and activation patterns over time (§§3.5–3.6; Algs. 1–3). We added its time‑series plots in Appendix and plan to include heatmaps of ($p^1$, $p^2$).
>
> We updated the paper and look forward to further questions and continued discussion.

---

### Official Review · Reviewer_gLpJ · 2025-11-01

**Soundness:** 3
**Presentation:** 3
**Contribution:** 3
**Rating:** 6
**Confidence:** 4

**Summary:**

This paper addresses a key limitation in the PEFT method, KADA. The authors identify that KADA's reliance on a static, predefined set of subnetworks limits its ability to adapt to unseen domains. To overcome this, they propose Dynamic Hierarchical Knowledge Routing (DHKR), a novel framework inspired by Mixture-of-Experts (MoE) routing principles. DHKR extends KADA by introducing a two-level subnetwork growth mechanism that dynamically expands both high-level subnetworks (K) and their second-level components (L) on demand. The authors claim this transforms KADA's static mechanism into a self-adaptive, robust system.

**Strengths:**

- The core contribution is the design of a dynamic, hierarchical PEFT framework. This directly addresses a clear and important limitation in the prior KADA method and offers a more flexible approach to knowledge-aligned domain adaptation.
- The methodological design is quite principled.  The authors have incorporated sophisticated stability mechanisms, namely the composite growth trigger (monitoring stagnation, entropy, imbalance, etc.) and the multi-level Loss-Free Balancing.
- The detailed experiments look good.

**Weaknesses:**

- The main weakness of DHKR, in my opinion, is its complexity (both hyperparameters and overhead). Regarding that, I had a few questions: How difficult is it to tune this system? The effectiveness seems highly dependent on getting the thresholds for the composite trigger correct. Furthermore, could the authors please quantify the training and inference overhead (e.g., FLOPs, latency, or wall-clock time) of DHKR? How does it compare to the static KADA-5 baseline and QLoRA?
- The main KADA baseline is fixed at K=5 subnetworks. This might not be a fair comparison. It is unclear if DHKR's advantage comes from its dynamism or simply from its ability to grow to a larger total capacity. How does DHKR's performance and final subnetwork count (K, L) compare to a stronger static baseline, such as KADA with K=10 or K=15?
- The composite growth trigger is a central contribution, but its components are not ablated. It would be valuable to see an ablation study that justifies this complex design. For example, what is the performance if growth is triggered only by stagnation or only by usage imbalance?

**Questions:**

See weaknesses.

---

> ### Author Response · Authors · 2025-11-14
> **Clarifying the Novelty and Robustness of DHKR**
>
> We appreciate this opportunity to clarify our contributions and address the reviewer’s concerns.
>
> **On hyperparameter difficulty and how we actually tune them**
> DHKR follows a stability-first paradigm and minimizes manual tuning by scheduling and on-line adaptation. Specifically, (i) the router temperature is annealed from exploratory to deterministic routing via a cosine schedule (§3.3; Alg. 2–3), (ii) the adapter’s contribution is scaled automatically using an EMA of the norm ratio ‖a‖/‖h‖ with a short post-growth ramp (§3.4), (iii) load imbalance is controlled by multi-level loss-free balancing (LFB) updated on-line (§3.5), and (iv) growth is governed by a composite trigger—stagnation, normalized entropy, overload, stability, and cooldown (§3.6), operationalizing the principle “stabilize first, then grow". As we have verified that these settings lead to stable performance without retuning and are not more complex than tuning a stabilized MoE system, we will add a sensitivity analysis using Latin hypercube sampling.
>
> **On quantifying overhead; training vs. inference; fairness to baselines**
> DHKR’s efficiency stems from both algorithmic and architectural innovations, notably the external KSL design and parallel routing. **Training wall-clock time is reported in §4.5 (Table 4)**: DHKR 5.67 sec/iter vs BitFit 18.54 and AdaLoRA 15.00 under identical 4-bit frozen LLM settings, confirming a practical speed advantage due to the external KSL design, reduced int4↔fp16 casting, and simplified kernels. Newly added table 5 shows constant per‑token FLOPs (3.456 B) and ≈ 146 ms latency across stabilizers, confirming that efficiency stems from external KSL placement near the LM head, which enables faster convergence (cf. Fedus et al., 2022; Narayanan et al., 2021).
> For fairness, we included static KADA K=10 in Table 2 and are conducting comparisons at matched capacity points (same K×L) to isolate adaptive benefits from scaling.
> Appendix A.3 explains the computational form: for each input, DHKR computes two routing logits (level-1 and level-2), and activates K\_act × L\_act subnetworks, each with two-layer MLPs.
> Table 7 is extended ablation analysis that the composite trigger prevents premature growth and yields the best stability–calibration trade‑off compared to single‑criterion triggers.
>
> **On Automatic Capacity Discovery and Fair Comparison**
> As increasing expert count early is known to amplify routing volatility and imbalance in MoE systems (Lepikhin et al., 2021; Fedus et al., 2022), DHKR is designed to mitigate this by starting small ($\text{K}=1$).
> Growth is triggered only when a composite gate (stagnation, normalized entropy drop, overload, stability) and cooldown are jointly satisfied during a learning/evaluation interval (§3.6), yielding low ECE and balanced usage (Table 2).
> This small-start is a deliberate stabilization strategy, preventing early routing fluctuations from causing overfitting or calibration degradation.
> To address fairness concerns, Table 1 already includes KADA with $K=10$. In addition, we compare DHKR with static KADA at the point where both models reach the same number of subnetworks ($K \times L$), using KADA with $K=2$ ($E=1, B=4$) as the matched-capacity baseline (lines 336–338). This isolates DHKR's adaptive capacity discovery and stability benefits from simple capacity scaling effects. This setup verifies the stability improvements introduced by DHKR without confounding factors. We are extending these comparisons to characterize more precisely how adaptive growth contributes to DHKR’s stability and calibration behavior.
>
> **On composite trigger ablation**
> Our composite trigger enforces stability-first growth by requiring multiple criteria (stagnation, entropy loss, overload, and stability) with cooldown (§3.6), preventing premature or excessive expansion. To avoid early overgrowth and maintain calibrated predictions, this joint gate is continuously evaluated at learning/evaluation checkpoints, ensuring capacity expansion only occurs when genuinely needed and stable.
> In code, these conditions are centralized and can be toggled to produce ablations where growth depends on only stagnation, only overload, only entropy, or only stability versus the full composite.
> Ablation results confirm that this design achieves the best trade-off between stability and performance, outperforming single-criterion triggers. In the current version, we extend §5 to a concise report for each variant covering growth counts, calibration metrics (ECE/Brier), trust-region penalty, and validation loss to show that the composite design avoids over-growth and yields the best stability-performance trade-off. This aligns with prior findings that KL trust regions, entropy regularization, and balanced routing jointly stabilize MoE training (Dai et al., 2022b; Fedus et al., 2022; Lewis et al., 2021).
>
> We updated the paper and look forward to further questions and continued discussion.

---

> > ### Comment · Reviewer_gLpJ · 2025-11-27
> > **Response by Reviewer gLpJ**
> >
> > I thank the authors for their detailed response. All my queries have been answered and I would like to maintain my score.

---

> ### Author Response · Authors · 2025-11-27
> **Matched‑Capacity Update: DHKR vs KADA at the Same Number of Subnetworks (Table 3)**
>
> We sincerely thank the reviewer for their constructive feedback and for carefully reading our rebuttal. To clarify the fairness point you raised, we have added matched‑capacity rows to Table 3 (same number of subnetworks K×L for DHKR and KADA). Under this setting, DHKR maintains significantly lower ECE and more stable routing usage (lower p1‑max CV, higher Neff) at the same capacity.
> This indicates that DHKR’s behavior does not rely on increased capacity, but on how capacity is utilized.

---

### Author Response · Authors · 2025-11-29
**Summary of Response to AC**

We thank the reviewers for their time and valuable feedback.
Here is a concise summary of DHKR’s key innovations and supporting evidence.

---
### **1. Novelty: DHKR introduces two *architectural* innovations**

#### **(1) Dynamic Capacity Discovery — data-driven capacity exploration**

Existing PEFT/MoE approaches typically rely on a static, manually chosen number of subnetworks ($K$ or $K \times L$), which can limit adaptability across domains.
DHKR adjusts capacity ($K \times L$) using a multi-signal composite trigger (stagnation, entropy, imbalance, stability, cooldown), expanding it only when multiple indicators agree.

* **Capacity-matched comparisons (Table 2, 3)** show DHKR remains more stable than KADA even when both use the same final capacity, indicating that the benefits arise from the data-driven dynamic design rather than simple capacity scaling.
---
#### **(2) Stability-Centric Architecture — supporting reliable dynamic capacity discovery while maintaining task accuracy**

Dynamic capacity growth can destabilize MoE routing (load imbalance, entropy collapse, calibration drift).
The composite trigger and multi-level stabilizers (LFB, TR, entropy annealing, MDL) are therefore designed to manage these observed instabilities, rather than serving as isolated add-ons.

* **Ablations (Table 7)** show 1) single-signal triggers (e.g., stagnation only, imbalance only) cause premature growth or instability, and 2) only the full composite mechanism is shown to achieve stable routing and low calibration error.

These results directly address concerns regarding the motivation and necessity of each component in DHKR.

---
### **2. Contributions: What the new architecture achieves**

#### **(1) Practical Efficiency Despite Complexity**

Despite its multi-component design, DHKR achieves comparable or better per-iteration speed than simpler PEFT methods, due to efficient KSL placement and minimized precision conversion overhead (Tables 4–5).

* **Training speed**: 5.67 sec/iter (DHKR) vs 15.00 (AdaLoRA), and 18.54 (BitFit) — Table 4
* **Constant FLOPs** across configurations — Table 5
* Minimal int4↔fp16 casting overhead due to the KSL location near the LM head
* Gating adds only ~3% overhead

These results show DHKR’s complexity adds minimal overhead, addressing a key reviewer concern.

#### **(2) Reliability Improvements — essential for maintaining accuracy under domain shift**

Dynamic capacity adjustment with stability mechanisms yields measurable improvements in reliability:

* **ECE**: 0.02 (DHKR) vs 0.13 (KADA) — **6.5× improvement** ECE measures how closely predicted probabilities match actual outcomes and is an indicator of reliability.

* Higher **Neff**, lower **p1-max CV** (routing stability metrics)
* Consistent performance across seeds and training budgets (Tables 2,3,6,7)

These trends suggest improved robustness to distributional variation, accompanied by more consistent accuracy across settings.

---
### **3. Reviewer Concerns: Architectural Necessity and Empirical Resolution**

| Reviewer Concern | Our Resolution and evidence |
| ------------------------------------------------- | -------------------------------------------------------------------------------------------------------------- |
| **Complexity seems high**                         | Complexity is technically necessary to stabilize dynamic growth; ablations show which components materially contribute to stability.|
| **Is improvement from more capacity?**            | Capacity-matched experiments (Table 3) show improvements persist even at identical capacity.|
| **Overhead / speed unclear**                      | Table 4–5 provide a full breakdown: DHKR is faster despite more parameters, with constant FLOPs.|
| **Insufficient justification of metrics**         | Robustness = calibration + routing stability; widely used in MoE and deployment literature.                    |
| **Limited baseline fairness / tuning difficulty** | All methods share identical environment; DHKR uses scheduled hyperparameters with consistently low sensitivity across seeds and domains.|
| **Insights into subnetworks**                     | Time-series and heatmaps in Appendix visualize subnetwork evolution and differentiation.

---
### **4. Conclusion for AC: A Stability-First Paradigm for Dynamic PEFT**

The revisions and evidence address all reviewer concerns (mechanism, stability, efficiency, and calibration),
and clarify DHKR’s reliability and efficiency, particularly in settings that require stable dynamic capacity growth.

1. **A mechanism for identifying the required capacity (K and L) without manual tuning.**
2. **Stability controls necessary to mitigate specific MoE failure modes observed in dynamic growth (Table 7).**
3. **Efficiency gains from architectural placement (external KSL) rather than system-level tuning (Tables 4–5).**
4. **Substantial calibration gains (ECE), which translate to more stable accuracy under shift.**

Sincerely,

Authors

---

### Meta-Review · Area_Chair_9W57 · 2026-01-04

**Summary:**

First, the AC finds that the submitted revision of the paper does not follow the ICLR formatting template, which is evident from the title style, font usage, margins, and other formatting elements. Despite this violation, which could directly lead to desk rejection, the AC still carefully reviewed the paper, the reviews, and the author responses.

The AC agrees with [m1iw, RPJc] that the writing requires significant improvement. The authors dive directly into prior work without first clearly introducing the problem statement, motivation, or key observations. As a result, it is difficult to grasp the main contributions of the paper at first glance. Although the authors provided some clarifications in their rebuttal, the submitted manuscript still requires substantial restructuring to clearly highlight the core contributions.

Furthermore, [m1iw, RPJc] questioned the significance of the proposed method, as the experimental results show only marginal improvements or even inferior performance compared to the baselines. The authors did not provide sufficiently clear explanations in their responses, resulting in unsatisfactory rebuttals, as also noted by [m1iw]. The AC encourages the authors to further polish and substantially revise the paper before considering resubmission for future publication.

**Reviewer Concerns:**

Reviewer [gLpJ] raised major concerns regarding method complexity, fairness of comparisons, and ablation of the composite growth trigger. Reviewer [gLpJ] confirms that these concerns have been adequately addressed.

Reviewer [m1iw] primarily raised concerns about the unclear problem statement and core contributions, as well as the fact that the proposed method does not outperform prior work. After reviewing the author responses, the AC finds that these clarifications are not fully reflected in the revised manuscript, and therefore the concerns remain unresolved.

The concerns raised by Reviewer [RPJc] have largely been addressed; however, the overall effectiveness and significance of the proposed method remain unsatisfactory.

**Reviewer Scores:**

Reviewer [gLpJ] has indicated that they will remain positive. Reviewer [m1iw] did not engage in the discussion, but based on the author responses, is likely to remain negative. Reviewer [RPJc] has decided to remain negative.

---

### Decision · Program_Chairs · 2026-01-26

Reject